# The impact of physical activity variety on physical activity participation

**Tyler M. Dregney** [1]*, **Chelsey Thul**[2], **Jennifer A. Linde** [3], **Beth A. Lewis**[2]

**1** Cancer Health Disparities T32 Training Program, Medical School and School of Public Health, University of Minnesota, Minneapolis, Minnesota, United States of America, **2** School of Kinesiology, College of Education and Human Development, University of Minnesota, Minneapolis, Minnesota, United States of America, **3** Division of Epidemiology & Community Health, School of Public Health, College of Education and Human Development, University of Minnesota, Minneapolis, Minnesota, United States of America

* dregn011@umn.edu

## Abstract

### Background and aims

Variety (i.e., multiple types of activities) may be effective for increasing physical activity (PA) based on previous research; however, research is needed to evaluate variety's impact on psychosocial variables.

### Methods

This exploratory study examined the effect of a home-based PA variety intervention on PA participation and psychosocial variables (motivation, psychological needs satisfaction, enjoyment, boredom, variety perception, PA feeling, self-efficacy, and affect) in an eight-week randomized intervention trial. Participants (n=47; mean age, 19.9 +/- 1.9; 75% female) were low-active, 18–25-year-old college students randomized to participate in the PA variety intervention or the consistency comparison condition. All participants received weekly individual counseling. The PA variety intervention received 14 unique high intensity interval training (HIIT) workouts with instructions to complete at least three different workouts per week, and the consistency comparison received one HIIT workout to complete at least three times per week. Priori comparisons and between groups analysis of covariance were used to examine findings.

### Results

Participants in the variety condition reported marginally significant more weekly moderate-to-vigorous PA ($p$=.072) over the course of the intervention and higher psychological needs satisfaction ($p$=.099) at four weeks relative to the comparison. The variety intervention condition reported significantly higher perceived autonomy ($p$=.013) within psychological needs at four weeks, PA feelings of tranquility ($p$=.005) at eight weeks, and PA self-efficacy ($p$=.025) at eight weeks relative to the comparison.

**Data availability statement:** All relevant data are within the manuscript and its Supporting Information files.

**Funding:** This study was supported by the National Cancer Institute (grant T32CA163184 to TMD).

**Competing interests:** The authors have declared that no competing interests exist.

## Conclusions

This exploratory study revealed there was preliminary evidence that variety may lead to improved psychological responses to PA among college-age individuals, although findings should be interpreted with caution given the use of marginal significance. Future studies should examine how a variety of different activities (e.g., cycling, tennis, group fitness classes) influence motivation and PA, in addition to including larger and more diverse samples. Practitioners should support clients' psychological needs and provide home-based PA options.

## Introduction

Sufficient physical activity (PA) leads to numerous health benefits including the prevention of type 2 diabetes, cardiovascular disease, cancer, hypertension, obesity, and osteoporosis [1–4]. Long-term PA participation is related to mental health benefits including improved cognitive function, reduced anxiety and depression risk, and improved quality of life [1,5]. However, PA decreases as individuals transition into early adulthood [6,7]. Despite potential benefits associated with PA, 66% of college-aged individuals in the U.S. do not meet PA guidelines [8].

Self-determination theory (SDT) provides a framework for how to understand PA behavior and increase PA participation [9,10]. SDT proposes that individuals must experience the basic psychological needs of perceived competence, autonomy, and relatedness to be intrinsically (internally) motivated, which promotes more effective pursuit of goal-directed behavior [10,11]. Researchers examining PA among young adults and college students often utilize SDT to guide interventions [12–15]. Findings suggest that SDT is appropriate for college students [12,13]. However, when interventions are not designed to address an individual's basic psychological needs, little impact is observed [14,15]. Initial research on variety in PA (i.e., multiple types of activities) indicates that variety may play a role in satisfying psychological needs and compensating for unsatisfied basic psychological needs, which can lead to increased intrinsic motivation for PA [16,17].

Preliminary research on variety indicates that increasing variety in PA leads to improvement in self-reported PA participation, motivation, and enjoyment [16–26]. Further, increasing one's variety in PA may decrease levels of boredom, as boredom in PA settings has been observed to lead to less PA [27]. Glaros and Janelle [20] examined the impact of a variety condition for adults in an eight-week intervention that had participants switch activities every two weeks. Findings indicated that the variety condition engaged in significantly more PA than participants who could choose any activity throughout the intervention. Additionally, the variety condition experienced significantly more enjoyment relative to the preferred and static (i.e., same activity for entire intervention) conditions. However, gaps and limitations remain for research that examines variety in PA.

Previous studies that examine the impact of variety in PA largely focus on children populations (18, 24–26). Previous studies have not examined variety's impact on PA

feeling, self-efficacy, or affective valence. Research that has examined variety in PA observed increased PA participation, motivation, and enjoyment as a result of variety in PA [16,20,21]; however, these studies did not include home-based interventions, objective measures of PA, or the exclusive use of cardiovascular fitness classes. One gap that persisted through previous research is the barriers to PA faced by college students. College students often mention not having enough time or energy and not having access to an activity facility as reasons for not being physically active [28]. Home-based PA is an effective method for college students to engage in enjoyable and sufficient PA [29]. A further limitation of previous research is the use of self-reported PA. PA measured via self-report can lead to bias in reporting and difficulty remembering PA participation, which skews findings [30]. This issue can be avoided through objective measurement of PA [31]. Finally, variety in PA has typically been examined in individual activities such as fitness training, stationary biking, or running [16,20]. However, cardiovascular fitness classes can also lead to improvements in fitness, perceived fitness, autonomy, and purpose in life for college students [32].

The present study addressed the following research gaps through: implementing an accessible, home-based variety PA intervention; applying objective measures to PA; utilizing cardiovascular fitness classes; and examining a wide breadth of psychosocial factors. Therefore, the primary purpose of this pilot study was to examine the feasibility and efficacy of a home-based PA variety intervention for low-active college students that aimed to increase PA participation and show positive effects on psychosocial variables in an eight-week randomized intervention trial.

As outlined by Bowen and colleagues [33], feasibility studies should examine acceptability and implementation and conduct limited efficacy testing. For this study, acceptability was achieved if participants completed a mean number of 75% of the motivational phone sessions and rated the intervention a mean number of six on a seven-point Likert scale. Implementation was achieved if at least 40 participants were recruited and at least 80% of the sample was retained at eight weeks. We determined that 40 participants would be adequate to determine if we could sustain a recruitment rate of 6–7 participants per month, which could inform a future trial seeking to recruit 200 participants over 2.5 years.

Regarding preliminary efficacy, it was hypothesized that participants randomized to the variety intervention would exhibit more PA participation at four and eight weeks relative to a comparison condition (described below). It was also hypothesized that participants randomized to the variety intervention would report greater increases in PA motivation, psychological need satisfaction, enjoyment, perception of variety, self-efficacy, affective valence and a decrease in boredom. Affective responses to workouts were also examined.

## Methods

### Overview of study

This study was a prospective, randomized controlled intervention pilot study conducted in the upper Midwest. Forty-seven low-active young adults were randomly assigned to either an eight-week PA variety intervention or a PA consistency comparison condition. PA was assessed via accelerometer (e.g., ActiGraph) and self-report at baseline and eight weeks. Psychosocial variables and affect were assessed via self-report at baseline, four, and eight weeks (i.e., post-intervention). This was an eight-week intervention given the effect previous PA variety interventions have had on PA participation and psychosocial variables [16,20]. This study received approval from the University of Minnesota's IRB and the study's ID is STUDY00017749. Written consent was obtained from all participants.

### Participants

Participants (n = 47) were recruited from September 19, 2023 through February 5, 2024 through psychology and kinesiology in-class announcements, email, and word of mouth. Inclusion criteria included the following: (1) reported engaging in no or low levels of PA [less than 90 minutes of moderate to vigorous intensity PA (MVPA) per week)], (2) capable of

completing a 30- minute session of PA, and (3) 18–25 years of age. Exclusion criteria included the following: (1) pregnancy, (2) cannot read in English, (3) no access to the internet, (4) healthcare providers had instructed them not to be active, and/or (5) any medical condition that would make PA unsafe or unwise.

## Procedures

Interested individuals contacted the primary investigator via email. The screening form was then emailed to interested individuals to determine eligibility. Following this, an in-person or online video conferencing meeting was scheduled to review detailed information about the study and the consent form. Individuals who remained eligible and interested completed the written consent form, and were then scheduled to pick up the accelerometer and sent the baseline questionnaires. Following the seven days of scheduled accelerometer wear time, participants returned the accelerometer and were randomized via a 1:1 ratio to either the variety or consistency condition using a random number generator in Microsoft Excel. Additionally, participants scheduled the first motivational phone session at this time.

Both conditions completed weekly, individual counseling sessions for the first four weeks, then every other week for the final four weeks (i.e., at six and eight weeks) of the eight- week intervention (the only component that differed between the conditions was the assignment of variety in PA or not). Sessions were scheduled at a time convenient for participants. To support participants' sense of competence, counseling sessions emphasized the importance of improvements experienced in PA. Participants recognized their improvement through support received during their sessions regarding their perceived ability to find success in workouts. To support participants' sense of autonomy, counseling sessions emphasized the importance of choosing the timing of a workout and the type of workout (for the variety condition) to be completed. Participant autonomy for PA was recognized through reminders that they were in charge of their behavior. To support participants' sense of relatedness, the counseling sessions reiterated the notion that the researchers were there to support them and their PA. The counseling sessions were held either via phone or Zoom (see Table 1 for individual session topics).

High intensity interval training (HIIT) workouts were provided to participants in both conditions. HIIT consists of bouts of PA interspersed with rest intervals. Examples of exercises included in the provided HIIT workouts were jump squats, high knees, plank variations, push-ups, jumping jacks, and other bodyweight exercises. Given this was a home-based study, HIIT workouts were presented to participants via a condition-specific website to allow participants to complete the

**Table 1. Timeline and content of counseling sessions.**

| Session | Strategy | Content |
|---|---|---|
| 1 | Overview of intervention | Describe the purpose of calls. Introduce the importance of variety/consistency in PA. Discuss barriers they currently face for PA. Schedule next session. |
| 2 | Autonomy | Discuss importance of feeling in control of actions. Remind participants they can choose the workouts (for variety condition) they do and when they do them. Prompt planning and time management. Schedule next session. |
| 3 | Competence | Discuss importance of feeling capable. Emphasize small improvements already made and how significant this accomplishment is. Provide instruction and encouragement. Schedule next session. |
| 4 | Relatedness | Discuss importance of having social support. Suggest strategies for gaining social support. Provide support to participants. Schedule next session. |
| 5 | Motivation and planning | Discuss importance of planning out workouts. Encourage planning that enhances motivation through timing or accountability. Address persisting barriers to PA. Provide assistance with planning. Schedule next session. |
| 6 | Enjoyment | Discuss importance of enjoyment for PA maintenance. Encourage participants to identify positive feelings and sensations they experience during activity. Discuss negative experiences in PA. Develop final goals. Give instructions and link for final assessments. |

Each of the strategies used were tailored to enhance the basic psychological needs of self-determination theory, motivation, and enjoyment for PA.

workouts in a space and time of their choosing. Despite individual counseling sessions not occurring each week of the intervention, participants in both conditions received weekly phone calls to log their PA from the previous week. All PA was logged regardless of if it was from the assigned website or an outside activity.

**Variety condition.** Upon completion of the randomization, participants in the variety condition were sent the link to their condition's website via email. The variety website contained 14 distinct HIIT videos. The videos were 30 minutes long and included a five-minute warm-up, 20-minute workout, and a five-minute cool down. The website also included workout plans on a PDF in a list format to allow participants to complete workouts without the need of a video. Video demonstrations of each exercise were available on the website. The exercise example videos included options for modifications to provide greater accessibility (e.g., regular push-ups, vs. knee push-ups, vs. wall push-ups). The variety condition was instructed to complete at least three different HIIT workouts per week and given encouragement to complete more than this if possible. Additionally, participants were informed at the beginning of the intervention they must choose different workouts throughout the week that were provided on the website. They did have the option to repeat workouts on a week-to- week basis but were instructed to not repeat any workouts during the same week.

**Consistency condition.** The consistency condition also received access to their website via email. The only difference between the variety and consistency condition's websites was the consistency condition's website contained just one HIIT video. Similar to the variety intervention, the consistency condition's video was 30 minutes long and included a five-minute warm-up, 20-minute workout, and a five-minute cool down. The website also contained the workout plan on a PDF and video exercise examples. The consistency condition was instructed to complete the provided HIIT workout three times per week and given encouragement to complete more than this if possible.

## Measures

**Primary aim: Feasibility.** Feasibility was assessed by examining acceptability and implementation. Level of acceptability was based on attendance at the motivational phone sessions and consumer satisfaction questionnaire responses (items were on a seven-point Likert scale). Implementation was based on recruitment and retention rates. We examined several variables to answer the question, "Can this study be done?", as is recommended by Eldridge and colleagues [34]. Specifically, we examined the number of eligible participants, percent of participants randomized from the eligible participant pool, follow-up rates, and time needed to collect data.

**Secondary dependent variables: PA participation and psychosocial variables.** PA was assessed via an accelerometer and a PA recall conducted by phone. PA was assessed at baseline and eight weeks via the ActiGraph. The ActiGraph is an electronic device worn on the right hip to identify motion, steps taken, energy expenditure, and time spent in different intensities of activity (light, moderate, hard, and very hard) [35,36]. Moderate, hard, and very hard intensity PA were combined to determine participants' MVPA. For ActiGraph data to be considered valid, participants needed to reach at least two days of valid wear time [37]. Psychosocial questionnaires included PA motivation, psychological needs satisfaction, enjoyment, boredom, PA feeling, self-efficacy, and affective valence. PA minutes per week were assessed during every week of the program using the 7-Day PA Recall Interview (PAR). The 7-day PAR [38] is considered the best measure for self-assessing PA and was used in this study to collect weekly self-reported PA participation during the counseling sessions [39].

The following psychosocial variables were assessed at baseline, four, and eight weeks. Motivation was assessed by two separate measures. The 19-item Behavioral Regulation in Exercise Questionnaire (BREQ-2) examined dimensions of motivation, as follows: amotivation (lack of motivation), external regulation (acting due to external guidance or pressure), introjected regulation (acting due to internal pressures), identified regulation (acting due to perceived importance of behavior), internal regulation (acting to achieve desired outcomes), and intrinsic motivation (acting due to internal rewards) [40]. The 30-item Motives for PA Measure-Revised (MPAM-R) examined motives for PA, including interest/enjoyment, competence, appearance, fitness, and social [41]. Validity has been established for both the BREQ-2 [35] and MPAM-R [42].

The 18-item Psychological Need Satisfaction in Exercise (PNSE) scale assessed satisfaction of psychological needs [43]. The PNSE has high reliability and is a commonly used measure for psychological needs [44].

The 18-item PA Enjoyment Scale (PACES) measured enjoyment [45]. This measure is reliable and valid [45]. The five-item Bored of Sports Scale (BOSS) assessed boredom in PA [27]. The BOSS is a reliable scale [27]. The Perceived Exercise Variety (PVE) questionnaire examined the perception of variety in PA [46]. The PVE is both valid and reliable [46]. The 12-item Exercise-Induced Feeling Inventory (EFI) measured PA feeling [47]. The EFI has been shown to correlate with related constructs [47]. The 10-item Exercise Self-Efficacy Scale (ESES) assessed confidence in one's ability to engage in PA [48]. This scale has acceptable reliability and validity among different populations [49]. The last psychosocial variable, affective valence (range of liking/dislike), was examined with the two-item Feeling Scale (FS) at one, four, and eight weeks during one of the weekly workouts [50]. The FS is reliable for assessing in-PA affect [46]. Full measures can be found in S1 File questionnaires.

## Data analysis

A priori comparison was conducted to analyze the total PA participation between the variety and consistency conditions. Between groups analysis of covariance (ANCOVA) was used to measure the effect of the intervention on the dependent variables at four and eight weeks. Baseline measures were included as covariates to control for baseline differences between conditions. A repeated measures ANOVA was conducted for each condition for the dependent variables to observe changes from baseline to eight weeks.

Additionally, given the small sample size, "marginal significance" was discussed for findings when $p < .10$ [51]. Data were analyzed using SPSS (v29.0) and Microsoft Excel (v16.84). The ActiGraph data was analyzed using ActiLife (v6.12.1). MVPA was the only ActiGraph data included in data analysis.

## Results

There were no between-group differences for any demographic variable or MVPA at baseline (Table 2). For baseline psychosocial variables, there were no differences between groups except for one subscale of the BREQ-2, identified regulation. Specifically, the variety intervention condition scored significantly higher on the identified regulation subscale of the BREQ-2 when compared to the consistency intervention $(p = .041)$.

### Feasibility

A flow chart detailing participant recruitment, screening, randomization, and retention is shown in Fig 1. Seventy-two potential participants were screened during the recruitment process, with 25 excluded due to lack of interest or deemed ineligible due to age and/or prior levels of PA. The retention rate was 92% for the variety intervention and 77% for the consistency comparison, with an 85% retention rate overall. There was no differential drop-out between conditions at four or eight weeks. Additionally, there were no differences in the demographic variables between those who dropped out (n = 7) and those who completed the study (n = 40).

Participants completed a mean number of 5.4 (SD = 0.14) of the six motivational phone sessions (98%). The consumer satisfaction mean score was 6.33 out of 7 (see Table 3). Participants in the variety intervention condition self-reported marginally significant higher use of workouts from the website, $f(1,41)=3.682$, $p = .076$, $d = .08$, and total workouts, $f(1,41)=3.072$, $p = .062$, $d = .08$, than the consistency comparison condition (Table 3). There were no differences between groups for participating in workouts not provided on the website or on the intervention satisfaction questionnaire.

### PA

There were no between-group differences for subjectively or objectively (see S1 Table) measured MVPA at four or eight weeks after controlling for baseline MVPA. The variety intervention condition reported marginally significant higher

**Table 2. Participant demographics by condition.**

| Characteristic | Total Sample (n-47) | Variety (n = 25) | Consistency (n = 22) | P-Value for Group Differences |
|---|---|---|---|---|
| Age | 19.94 (1.90) | 19.92 (1.98) | 19.95 (1.86) | .951 |
| Gender (%) | | | | .620 |
| Female | 79% | 76% | 82% | |
| Male | 19% | 20% | 18% | |
| Non-conforming | 2% | 4% | 0% | |
| Race (%) | | | | .280 |
| Asian | 40% | 36% | 45% | |
| Black/African American | 11% | 8% | 14% | |
| White | 45% | 52% | 36% | |
| Other | 4% | 4% | 5% | |
| Ethnicity (%) | | | | .750 |
| Hispanic | 11% | 12% | 9% | |
| Not Hispanic or Latino | 89% | 88% | 91% | |
| Baseline Objective MVPA | 251.19 (127.05) | 258.93 (123.80) | 241.35 (136.47) | .739 |
| Baseline Subjective MVPA | 73.72 (85.36) | 80.00 (92.11) | 66.50 (78.63) | .610 |

MVPA = Moderate-to-vigorous physical activity.

average weekly self-reported MVPA than the consistency condition, $f(1,40)=3.416$, $p = .072$, $d = .08$, after controlling for baseline MVPA (see S2 Table).

Based on the repeated measures analysis, participants in the variety intervention condition self-reported significant increases for MVPA from baseline to eight weeks, $f(1,22)=18.678$, $p < .001$, $d = .46$, with no changes reported for objectively measured PA. Participants in the consistency comparison condition also reported significantly increased MVPA from baseline to eight weeks based on the repeated measures analysis, $f(1,16)=6.202$, $p = .024$, $d = .28$.

## Psychosocial variables

There were no differences between conditions for motivation (see S3 and S4 tables), enjoyment (Table S5), or boredom (Table S6) at four or eight weeks. The variety intervention reported significantly higher satisfaction for autonomy (Table S7) at four weeks [$f(1,38)=6.768$, $p = .013$, $d = .15$], perception of variety in PA (Table S8) at four [$f(1,38)=0.778$, $p = .011$, $d = .16$] and eight weeks [$f(1,37)=8.443$, $p = .006$, $d = .19$], tranquility (Table S9) at eight weeks [$f(1,37)=8.783$, $p = .005$, $d = .19$], and PA self-efficacy (Table S10) at eight weeks [$f(1,37)=5.439$, $p = .025$, $d = .13$] relative to the consistency comparison condition. The variety intervention reported marginally significant higher basic psychological needs satisfaction at four weeks [$f(1,38)=2.859$, $p = .099$, $d = .07$] relative to the consistency comparison condition (Table S7).

Based on repeated measures analysis, the variety intervention condition reported a significant increase in intrinsic motivation [$f(1,22)=10.904$, $p = .003$, $d = .33$], identified regulation [$f(1,22)=13.144$, $p < .001$, $d = .37$], motives of enjoyment [$f(1,22)=12.523$, $p = .002$, $d = .36$], motives of competence [$f(1,22)=11.221$, $p = .003$, $d = .34$], perceived competence [$f(1,22)=11.221$, $p = .001$, $d = .39$], perceived autonomy [$f(1,22)=5.107$, $p = .034$, $d = .19$], psychological need satisfaction [$f(1,22)=2.652$, $p < .001$, $d = .04$], enjoyment [$f(1,22)=24.329$, $p < .001$, $d = 53$], perception of variety in PA [$f(1,22)=32.088$, $p < .001$], engagement [$f(1,22)=9.781$, $p = .005$, $d = .31$], revitalization [$f(1,22)=29.776$, $p < .001$, $d = .58$], tranquility [$f(1,22)=21.894$, $p < .001$; $d = .50$], PA self-efficacy [$f(1,22)=15.271$, $p < .001$, $d = .41$], and decreases in boredom [$f(1,22)=10.252$, $p = .004$, $d = .32$] and physical exhaustion [$f(1,22)=11.535$, $p = .003$, $d = .20$] from baseline to eight weeks. The consistency comparison condition reported a significant increase in amotivation [$f(1,16)=8.157$, $p = .011$, $d = .34$],

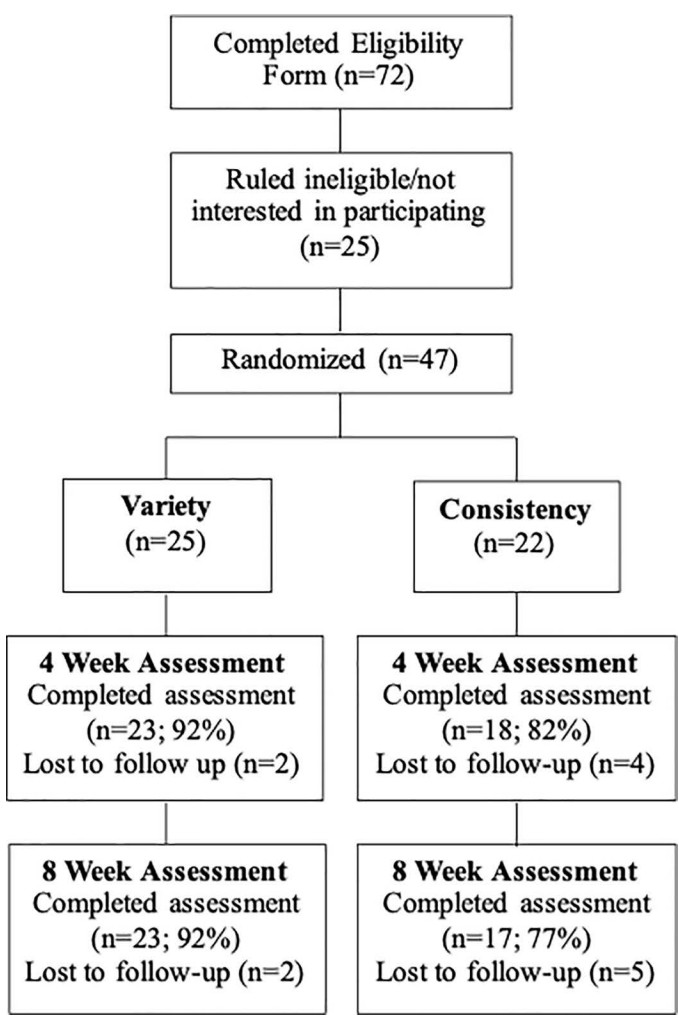

**Fig 1. Variety flow chart.**

**Table 3. Means and standard deviations for intervention adherence and satisfaction by conditions.**

| Variable | Total | | Variety | | Consistency | |
|---|---|---|---|---|---|---|
| | M | (SD) | M | (SD) | M | (SD) |
| Website Workouts per Week | 2.17 | (1.26) | 2.48[a] | (1.16) | 1.76 | (1.30) |
| Additional Workouts per Week | 0.70 | (0.70) | 0.66 | (0.71) | 0.73 | (0.69) |
| Total Workouts per Week | 3.14 | (0.77) | 3.33[a] | (0.72) | 2.93 | (0.79) |
| Intervention Satisfaction | 6.33 | (0.72) | 6.30 | (0.82) | 6.37 | (0.60) |

[a]Difference is marginally significant at $p < 0.10$; *Difference is significant at $p < .05$; **Difference is significant at $p < .01$; ***Difference is significant at $p < .001$; Standard deviations are listed in parentheses.

identified regulation [$f(1,16)=5.755$, $p=.029$, $d=.27$], motives of enjoyment [$f(1,22)=11.221$, $p=.039$, $d=.24$], and enjoyment [$f(1,16)=6.952$, $p=.018$, $d=.30$] from baseline to eight weeks.

There were no between group differences for pleasurable or enjoyment affect at any time point (Table S11). The variety intervention did report an increase in pleasurable affect before [$f(1,22)=6.823$, $p=.016$, $d=24$], during [$f(1,22)=9.307$, $p=.006$, $d=.30$], and after [$f(1,22)=7.560$, $p=.012$, $d=.26$] the workout from baseline to eight weeks. The variety intervention also reported an increase in enjoyment affect before [$f(1,22)=9.722$, $p=.005$, $d=.31$] and during [$f(1,22)=7.057$, $p=.014$, $d=.24$] the workout from baseline to eight weeks. The consistency comparison condition experienced no changes in either affect from baseline to eight weeks.

## Discussion

This home-based PA variety intervention appears feasible based on meeting the acceptability and implementation parameters. Specifically, the recruitment goal was met, as 47 participants were randomized. The retention rate (85%) was higher than the 80% retention goal. Additionally, participants completed a mean number of 98% of the motivational phone sessions, which surpassed the goal of 75%. The mean rating for participant satisfaction in the variety condition was 6.30 while the consistency comparison was 6.37, both exceeding the goal of six on the seven-point scale.

Regarding efficacy, this study suggests that a variety intervention may be efficacious for increasing PA relative to a consistency comparison; however, the results are inconclusive. Consistent with the hypothesis and previous studies [16,20], the variety intervention condition reported more average weekly self-reported MVPA minutes at the counseling sessions than the consistency comparison condition. Additionally, Chiang and colleagues [52] examined how a home-based PA program that allowed participants to choose their workouts from impacted PA participation. Participants who were in the home-based choice intervention reported higher PA participation relative to the control. It is possible having more choice also influenced the findings of the present study. This is further supported as variety participants also reported completing more HIIT sessions from the assigned website than the control participants.

Despite the variety intervention condition reporting higher average weekly self-reported MVPA at the weekly counseling sessions, there were no between-group differences for objectively or subjectively measured MVPA at the four- or eight-week assessment sessions after controlling for baseline. A potential confounding variable that may explain the lack of MVPA differences between groups at four and eight weeks specifically is that the consistency comparison condition was an active control, meaning the weekly counseling sessions could have led to increases in their PA levels. The sessions were designed to enhance participants' basic psychological needs through supporting competence, autonomy, and relatedness. When satisfied, these basic psychological needs lead individuals to gain a sense of intrinsic motivation [10]. Since both conditions received these sessions and experienced increases in PA over time, it is possible that all participants felt compelled to complete the workouts provided to them. This is supported by the fact that both conditions significantly increased their PA levels from baseline to eight weeks. Regardless, the present study suggests that variety may lead to increased PA participation.

Contrary to the hypothesis and previous research, there were no between-group differences for motivation at four or eight weeks; however, participants in the variety intervention condition reported significantly increased intrinsic motivation from baseline to eight weeks, which does align with previous research [16,17]. The lack of differences between groups for motivation can also potentially be attributed to the weekly counseling sessions that were designed to enhance these forms of motivation specifically for both groups. Further, the study duration may not have been sufficient for the intervention design to significantly influence participant motivation.

Consistent with the hypothesis, the variety intervention condition reported higher levels of psychological needs satisfaction and autonomy at four weeks. However, there were no between-group differences for psychological needs satisfaction at eight weeks. The variety intervention condition increased their perceived competence, autonomy, and satisfaction of basic psychological needs from baseline to eight weeks. The variety intervention condition reporting an increase in both

satisfaction of basic psychological needs and intrinsic motivation from baseline to eight weeks aligns with SDT as PA also increased for this condition during this time [10,11,53].

Contrary to the hypothesis and previous research [20,21], there were no between-group differences for enjoyment or boredom at four or eight weeks. Glaros and Janelle [20] conducted an eight-week intervention and found increased enjoyment in a variety condition as compared to a static condition. However, the variety condition changed the type of activity they were doing every two weeks. It is possible that if participants in the present study changed activity every two weeks rather than completing the same mix of HIIT workouts throughout the entire intervention, an increase in enjoyment may have been observed. Although there were no differences between groups, the variety intervention condition did report significantly decreased boredom from baseline to eight weeks. Previous research has not examined the relationship between boredom and variety in PA. However, Wolff and colleagues [27] observed that low levels of boredom are associated with increased PA, which aligns with the increase in PA from baseline to eight weeks for the intervention condition.

Consistent with the hypothesis, participants in the variety condition reported significantly higher perceived variety at four weeks and higher perceived variety, self-efficacy, and tranquility at eight weeks relative to the consistency condition. There were no between-group differences for engagement, revitalization, physical exhaustion, or affective valence at any time point. In spite of these similarities between groups, the variety condition reported a significant increase in perceived variety of PA, self-efficacy, engagement, revitalization, and tranquility, while also reporting a decrease in physical exhaustion from baseline to eight weeks. Previously, Sylvester and colleagues [46] found that perceived variety was positively related to PA participation. No previous research has examined variety's impact on self-efficacy, engagement, revitalization, tranquility, physical exhaustion, or affective valence.

There were several limitations of this study. First, participants were primarily female and non-Hispanic, which limits the generalizability of the results. Second, due to this being a pilot study examining the feasibility and acceptability of variety in PA, a small sample size (n = 47) was recruited. The nature of the study also portended that no power analysis would be ran, which lowered the ability to detect between-group differences, which resulted in the discussion "marginal significance" to advise that findings be interpreted with caution and conclusions not be overstated. Third, despite participants reporting that they were active less than 90 minutes per week during the screening process, some were above this threshold during the baseline data collection.

Fourth, participants in both conditions engaged in physical activities beyond the provided workouts, which may have contributed to the lack of differences between groups. Fifth, both conditions receiving the weekly counseling session may have acted as a confounder, which also may have contributed to the lack of differences between groups. Lastly, there was a large gap in PA participation when comparing the subjective and objective results. This gap may be due to the small sample of objectively measured data, due to low compliance with wearing the monitor and lost data due to malfunctioning of the ActiGraph [54]. Despite research suggesting two days of validated wear time of the ActiGraph may be sufficient [37,55], not all participants reach this threshold. Further, Trost and colleagues [56] observed the minimum accelerometer wear time for reliable data to be 10 or more hours per day for at least four days. Although supported by previous research, it is possible the two-day minimum wear time led to inaccurate results.

The present study had several strengths. This study was the first to examine variety in a home-based intervention. PA variety interventions thus far have not been home-based, despite previous research suggesting that home-based PA interventions are efficacious in college populations [29,57]. This study also was the first to examine variety exclusively within cardiovascular fitness classes rather than individual activities (i.e., running, rowing, and cycling) and resistance training [16,20]. This study was also the first to examine variety's impact on PA feeling, self-efficacy, and affective valence. Finally, this study's sample was relatively diverse when compared to the overall population of the recruitment area.

The home-based, PA variety intervention appears feasible based on recruitment, retention, satisfaction, and adherence. There was some evidence for efficacy as the variety intervention had higher PA, self-efficacy, tranquility, autonomy, and psychological need satisfaction, although some of these findings were marginal. Future research should

consider implementation of variety in PA using novel strategies, new populations, and longer interventions. Specifically, studies should examine variety through variation of types of PA (i.e., swimming, sports, rock climbing etc.) in the same program. Studies should examine the application of variety to fulfill PA guidelines of at least 150 minutes of moderate or 75 minutes of vigorous cardiovascular PA, in addition to two sessions of strength training per week [2]. Particularly, studies should examine the effect of variety in PA among older adults, non-college educated individuals, males, and Hispanic individuals, as research is lacking in this area among these groups [58]. Finally, studies should examine longer interventions of various lengths, and post-intervention follow-ups to determine the long-term effects of a PA variety intervention.

There was some evidence that variety may be important for encouraging PA participation; however, some of the results were marginal and there were inconsistencies across measures and timepoints. Despite the inconsistent findings, this study has possible implications for practitioners. Specifically, practitioners should consider providing a variety of activities to clients to enhance one's self-efficacy to be physically active. Providing variety through choice in individual exercises and workouts may enhance perceived autonomy, which is a key component in supporting and building intrinsic motivation. Additionally, practitioners should consult and support clients when providing home-based PA to ensure psychological needs are being satisfied. Both strategies proved effective in improving individual's PA participation and motivation for both conditions. Finally, practitioners should consider client preferences when determining PA routines and addressing PA barriers. Future studies should address the limitations of the present study by examining variety through novel methods, recruiting larger and more diverse samples, applying different intervention lengths and follow-ups, and utilizing objective measures of PA.

## Supporting information

**S1 Table. Means and standard deviations for weekly minutes of subjectively and objectively measured MVPA by condition.**
(DOCX)

**S2 Table. Means and standard deviations for average weekly MVPA of the intervention by condition.**
(DOCX)

**S3 Table. Means and standard deviations for BREQ-2 by condition.**
(DOCX)

**S4 Table. Means and standard deviations for MPAM-R by condition.**
(DOCX)

**S5 Table. Means and standard deviations for PACES by condition.**
(DOCX)

**S6 Table. Means and standard deviations for BOSS by condition.**
(DOCX)

**S7 Table. Means and standard deviations for PNSE by condition.**
(DOCX)

**S8 Table. Means and standard deviations for PVE by condition.**
(DOCX)

**S9 Table. Means and standard deviations for EFI by condition.**
(DOCX)

**S10 Table. Means and standard deviations for ESES by condition.**
(DOCX)

**S11 Table. Means and standard deviations for FS by condition.**
(DOCX)

**S1 File. Questionnaires.**
(DOCX)

**S2 File. Data.**
(XLSX)

## Acknowledgments

We would like to thank Kaele Ojeda and Chloe Nelson for their assistance with this project. We would also like to thank all the participants in this study. Their commitment and the time they spent dedicated to actively engaging in this study was especially appreciated.

## Author contributions

**Conceptualization:** Tyler M. Dregney, Chelsey Thul, Jennifer A. Linde, Beth A. Lewis.

**Data curation:** Tyler M. Dregney.

**Formal analysis:** Tyler M. Dregney.

**Investigation:** Tyler M. Dregney.

**Methodology:** Tyler M. Dregney, Beth A. Lewis.

**Project administration:** Tyler M. Dregney.

**Software:** Tyler M. Dregney.

**Supervision:** Tyler M. Dregney, Beth A. Lewis.

**Validation:** Tyler M. Dregney.

**Visualization:** Tyler M. Dregney.

**Writing – original draft:** Tyler M. Dregney.

**Writing – review & editing:** Chelsey Thul, Jennifer A. Linde, Beth A. Lewis.

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
