## [Decision Letter · Decision Letter 0]

28 Nov 2024

PONE-D-24-49440The Impact of Physical Activity Variety on Physical Activity ParticipationPLOS ONE

Dear Dr. Dregney,

Thank you for submitting your manuscript to PLOS ONE. After careful consideration, we feel that it has merit but does not fully meet PLOS ONE’s publication criteria as it currently stands. Therefore, we invite you to submit a revised version of the manuscript that addresses the points raised during the review process.

We look forward to receiving your revised manuscript.

Kind regards,

Henri Tilga, PhD

Academic Editor

PLOS ONE

Reviewers' comments:

Reviewer's Responses to Questions

**Comments to the Author**

1. Is the manuscript technically sound, and do the data support the conclusions?

Reviewer #1: Partly

Reviewer #2: Partly

2. Has the statistical analysis been performed appropriately and rigorously? 

Reviewer #1: Yes

Reviewer #2: I Don't Know

3. Have the authors made all data underlying the findings in their manuscript fully available?

Reviewer #1: Yes

Reviewer #2: Yes

4. Is the manuscript presented in an intelligible fashion and written in standard English?

Reviewer #1: Yes

Reviewer #2: Yes

5. Review Comments to the Author

Reviewer #1: The manuscript investigates the impact of physical activity variety on participation levels and associated psychosocial variables among college-aged individuals. Address the methodological limitations and enhance the intervention’s scope to strengthen the study’s validity and applicability.

Major Weaknesses:

1.The sample size (n=47) is small, limiting the ability to detect significant between-group differences.

2.Despite promising trends, key outcomes such as moderate-to-vigorous physical activity (MVPA) and enjoyment lack statistically significant differences between conditions.

3.Marginal significance in several findings (e.g., autonomy, self-efficacy) raises questions about the robustness of conclusions.

4.The sample is skewed toward female, non-Hispanic participants, limiting applicability to broader populations.

5. The variety intervention was restricted to High-Intensity Interval Training (HIIT) videos, which may not appeal to participants with diverse activity preferences. A broader variety (e.g., yoga, cycling) might yield more compelling results.

Minor Concerns:

1. Discrepancies between subjective and objective MVPA underscore potential biases in self-reporting, which should be addressed in future studies.

2. Measures such as boredom and enjoyment lack significant between-group differences, contrary to prior studies. This may indicate a need for adjustments in intervention design (e.g., changing activities more frequently).

Reviewer #2: Comments for the author

This manuscript, entitled 'The Impact of Physical Activity Variety on Physical Activity Participation,' addresses an important and interesting research issue that provides a reference for promoting physical activity among college-aged individuals. However, several descriptions in the manuscript are unclear and require more detail. In addition, there are too many tables. Please combine them or focus on the most important three to four tables, relegating all others to supplemental tables..

1. In the abstract, it is suggested that the first sentence be rewritten for completion: 'Research indicates that variety (i.e., multiple types of activities) may be effective for increasing physical activity due to...?' Please provide a critical reason to complete the sentence."

2. In the abstract, what is the physical activity variety intervention? Please provide more details.

3. In the Methods section of the abstract, please describe the major outcome indicators investigated in the study and provide details about the primary statistical methods used.

4. In the Conclusion section of the abstract, it is highlighted that 'there was some evidence that variety may lead to increased physical activity...' However, you mentioned that the study results indicated 'Participants in the variety condition reported a marginally significant increase in weekly moderate-to-vigorous physical activity (p = .072).' Please revise the conclusion to reflect that this finding is not statistically significant.

5. On page 3, please change the description from 'type II diabetes' to the updated term 'type 2 diabetes'.

6. On page 3, in the second paragraph, you described that 'Researchers examining physical activity (PA) among young adults and college students often utilize Self-Determination Theory (SDT) to guide interventions.' If this statement is based on previous evidence, using the word 'will' might be inappropriate. Additionally, please provide more details about the examples of PA variety in the following sentence: 'Initial research on PA variety (i.e., multiple types of activities)...'.

7. Regarding the PA variety intervention, you mentioned that limited literature was found. However, there are some studies applied interventions of patient-preferred PA types. Please include the following references and discuss more details about using patient-preferred PA types.

Chiang, S. L., Shen, C. L., Chen, L. C., Lo, Y. P., Lin, C. H., & Lin, C. H.* (2020). Effectiveness of a home-based, telehealth exercise training program for patients with cardiometabolic multimorbidity: A randomized controlled trial. Journal of Cardiovascular Nursing, 35 (5), 491–501.

Chiang, S. L., Shen, C. L., Lee, M. S., Lin, C. H., & Lin, C. H.* (2023). Effectiveness of a 12-week tele-exercise training program on cardiorespiratory fitness and heart rate recovery in patients with cardiometabolic multimorbidity: A randomized controlled trial. Worldviews on Evidence-Based Nursing, 20(4), 339-350.: https://doi:10.1111/wvn.12607

Lai, C. Y., Lin, C. H., Chao, T. C., Lin, C. H., Chang, C. C., Huang, C. Y., & Chiang, S. L. (2024). Effectiveness of a 12-week tele-exercise training program in patients with long COVID. Annals of Physical and Rehabilitation Medicine, 67(5), 101853. https://doi.org/https://doi.org/10.1016/j.rehab.2024.101853.

8. On page 4, please expand the discussion in the sentence: 'Furthermore, few studies have examined the efficacy of PA interventions focusing on variety.' Additionally, in the following reference, it is also reported that there was a significant increase in PA self-efficacy.

Lai, C. Y., Lin, C. H., Chao, T. C., Lin, C. H., Chang, C. C., Huang, C. Y., & Chiang, S. L. (2024). Effectiveness of a 12-week tele-exercise training program in patients with long COVID. Annals of Physical and Rehabilitation Medicine, 67(5), 101853. https://doi.org/https://doi.org/10.1016/j.rehab.2024.101853.

9. On page 4, you mentioned, 'Home-based PA is an effective method for college students to engage in enjoyable and sufficient PA,' which I agree with. However, in my opinion, most college students engage in home-based PA rather than using activity facilities. Therefore, home-based PA might not be the issue. Alternatively, you could highlight the prevalence of engaging in activity facilities among college students. Please clarify.

10. Please provide the sample size estimation.

11. Please move the 'Measure' section to the paragraph before 'Data Analysis' and after the 'Intervention' section.

12. PA data was collected using the ActiGraph. Please provide more details about the measurement tool, especially its validity and reliability.

13. On page 9, the paragraph states: 'High-intensity interval training (HIIT) workouts were provided to participants in both conditions....' What are the details of the exercise prescription, including frequency, time, and duration?

14. In the Results section, there are too many tables. Please combine them or focus on the most important three to four tables, and relegate all other tables to supplemental materials.

15. In the Discussion section, the major findings should be discussed, especially in comparison to the previous studies mentioned above.

6. PLOS authors have the option to publish the peer review history of their article (what does this mean? ). If published, this will include your full peer review and any attached files.

**Do you want your identity to be public for this peer review?** For information about this choice, including consent withdrawal, please see our Privacy Policy .

Reviewer #1: No

Reviewer #2: **Yes: ** Chia-Huei Lin

---

## [Author Response · Author response to Decision Letter 0]

9 Dec 2024

Academic Editor:

a. The title page now matches the required format and all necessary information is provided. Additionally, headers now match the journal’s preferred format. Finally, the figure file has been converted to a TIFF file.

a. Thank you for catching this. A statement has been added at the end of the first paragraph of the methods section (see final paragraph of page 6) to address these concerns.

a. Fig. 1 now has a separate caption that includes the title on page 12 of the manuscript.

Reviewer A

Major concerns:

1. The sample size (n=47) is small, limiting the ability to detect significant between-group differences.

a. Thank you for raising this. We agree with this limitation. We have listed sample size as one of the limitations of this study and called for future studies to include larger samples in the discussion section (see paragraph 1 on page 19 and paragraph 2 on page 20).

2. Despite promising trends, key outcomes such as moderate-to-vigorous physical activity (MVPA) and enjoyment lack statistically significant differences between conditions.

a. We agree this is a fair concern. We have listed this as a limitation in the discussion section (see paragraph 1 on page 19).

3. Marginal significance in several findings (e.g., autonomy, self-efficacy) raises questions about the robustness of conclusions.

a. We recognize this as a limitation of the manuscript. Therefore, this limitation is discussed in the analysis (see paragraph 1 on page 11) and discussion sections (see paragraph 1 on page 19).

4. The sample is skewed toward female, non-Hispanic participants, limiting applicability to broader populations.

a. Thank you for pointing this out. We have listed this as a limitation and called for future studies to include more diverse samples in the discussion section (see paragraph 1 on page 19 and paragraph 2 on page 20).

5. The variety intervention was restricted to High-Intensity Interval Training (HIIT) videos, which may not appeal to participants with diverse activity preferences. A broader variety (e.g., yoga, cycling) might yield more compelling results.

a. We also view this as a limitation to this study. Therefore, we discussed this topic in the limitation section of the discussion section (see paragraph 1 on page 19)

Minor concerns:

1. Discrepancies between subjective and objective MVPA underscore potential biases in self-reporting, which should be addressed in future studies.

b. We recognize this as a limitation. Due to this, we have listed this as a limitation and called for future studies to include more diverse samples in the discussion section (see paragraph 1 on page 19 and paragraph 2 on page 20).

2. Measures such as boredom and enjoyment lack significant between-group differences, contrary to prior studies. This may indicate a need for adjustments in intervention design (e.g., changing activities more frequently).

a. Thank you for raising this point. We have discussed the potential of the situation described here in the discussion section (see paragraph 1 on page 18).

Reviewer B

1. In the abstract, it is suggested that the first sentence be rewritten for completion: 'Research indicates that variety (i.e., multiple types of activities) may be effective for increasing physical activity due to...?' Please provide a critical reason to complete the sentence."

a. Thank you for pointing this out. We have restructured the first sentence to reflect this recommendation. It now states that we believe a variety intervention is effective due to existing research (see abstract).

2. In the abstract, what is the physical activity variety intervention? Please provide more details.

a. This is a very good point. We have added further explanation to what the variety and consistency groups received for content and instructions (see abstract).

3. In the Methods section of the abstract, please describe the major outcome indicators investigated in the study and provide details about the primary statistical methods used.

a. Thank you for catching this. The major outcomes are listed in the first sentence of the abstract methods section and the primary statistical methods used are noted in the final sentence of the section (see abstract).

4. In the Conclusion section of the abstract, it is highlighted that 'there was some evidence that variety may lead to increased physical activity...' However, you mentioned that the study results indicated 'Participants in the variety condition reported a marginally significant increase in weekly moderate-to-vigorous physical activity (p = .072).' Please revise the conclusion to reflect that this finding is not statistically significant.

a. We agree with this point. This part of the conclusion has been removed (see abstract).

5. 5. On page 3, please change the description from 'type II diabetes' to the updated term 'type 2 diabetes'.

a. Thank you for catching this. This change has been made (see paragraph 1, page 3).

6. On page 3, in the second paragraph, you described that 'Researchers examining physical activity (PA) among young adults and college students often utilize Self-Determination Theory (SDT) to guide interventions.' If this statement is based on previous evidence, using the word 'will' might be inappropriate. Additionally, please provide more details about the examples of PA variety in the following sentence: 'Initial research on PA variety (i.e., multiple types of activities)...'.

a. Good point. The word “will has been removed. Additionally, an example of a specific intervention has been added to the third paragraph (see paragraph 2 on page 3).

7. Regarding the PA variety intervention, you mentioned that limited literature was found. However, there are some studies applied interventions of patient-preferred PA types. Please include the following references and discuss more details about using patient-preferred PA types.

Chiang, S. L., Shen, C. L., Chen, L. C., Lo, Y. P., Lin, C. H., & Lin, C. H.* (2020). Effectiveness of a home-based, telehealth exercise training program for patients with cardiometabolic multimorbidity: A randomized controlled trial. Journal of Cardiovascular Nursing, 35 (5), 491–501.

Chiang, S. L., Shen, C. L., Lee, M. S., Lin, C. H., & Lin, C. H.* (2023). Effectiveness of a 12-week tele-exercise training program on cardiorespiratory fitness and heart rate recovery in patients with cardiometabolic multimorbidity: A randomized controlled trial. Worldviews on Evidence-Based Nursing, 20(4), 339-350.: https://doi:10.1111/wvn.12607

Lai, C. Y., Lin, C. H., Chao, T. C., Lin, C. H., Chang, C. C., Huang, C. Y., & Chiang, S. L. (2024). Effectiveness of a 12-week tele-exercise training program in patients with long COVID. Annals of Physical and Rehabilitation Medicine, 67(5), 101853. https://doi.org/https://doi.org/10.1016/j.rehab.2024.101853.

a. Thank you sending these our way. We have elected to include these studies in the discussion section. Although participants in these studies had access to multiple types of physical activity, they were not specifically instructed to complete different exercises/workouts. Therefore, we felt this would contribute to more appropriately to the variety narrative in the discussion (see paragraph 2, page 16).

8. On page 4, please expand the discussion in the sentence: 'Furthermore, few studies have examined the efficacy of PA interventions focusing on variety.' Additionally, in the following reference, it is also reported that there was a significant increase in PA self-efficacy.

Lai, C. Y., Lin, C. H., Chao, T. C., Lin, C. H., Chang, C. C., Huang, C. Y., & Chiang, S. L. (2024). Effectiveness of a 12-week tele-exercise training program in patients with long COVID. Annals of Physical and Rehabilitation Medicine, 67(5), 101853. https://doi.org/https://doi.org/10.1016/j.rehab.2024.101853.

a. Thank you for pointing this out. We have elected not to include this study in the introduction. Although participants in this study had access to multiple types of physical activity, they were not specifically instructed to complete different exercises/workouts. Therefore, we do not feel this would contribute to the variety narrative in the introduction.

9. On page 4, you mentioned, 'Home-based PA is an effective method for college students to engage in enjoyable and sufficient PA,' which I agree with. However, in my opinion, most college students engage in home-based PA rather than using activity facilities. Therefore, home-based PA might not be the issue. Alternatively, you could highlight the prevalence of engaging in activity facilities among college students. Please clarify.

a. This is a very good point and something we thought would be the case as well. However, research suggests that college student view distance to an activity facility as a significant barrier (cited in this paragraph with citation #24). Therefore, we left the section as is.

10. Please provide the sample size estimation.

a. Thank you for catching this. A description for while 40 participants was chosen is now included in the introduction (see paragraph 2, page 5).

11. Please move the 'Measure' section to the paragraph before 'Data Analysis' and after the 'Intervention' section.

a. We appreciate this suggestion. This change has been made.

12. PA data was collected using the ActiGraph. Please provide more details about the measurement tool, especially its validity and reliability.

a. This is a very good idea. We have added more details regrding the ActiGraph and a sentence establishing validity and reliability (see paragraph 2, page 9).

13. On page 9, the paragraph states: 'High-intensity interval training (HIIT) workouts were provided to participants in both conditions....' What are the details of the exercise prescription, including frequency, time, and duration?

a. The details for the variety intervention, including the details of the exercises are discussed in the variety section of the procedures (see paragraph 1, page 8).

14. In the Results section, there are too many tables. Please combine them or focus on the most important three to four tables, and relegate all other tables to supplemental materials.

a. Thank you for pointing this out and we share this concern. We have combined the PA tables into one (see top of page 14). The psychosocial tables have been relegated to supplemental materials as requested.

15. In the Discussion section, the major findings should be discussed, especially in comparison to the previous studies mentioned above.

a. Thank you for pointing this out. We believe this has been addressed through the incorporation of one of the studies you suggested (see paragraph 2, page 16). The citation is number 48.

---

## [Decision Letter · Decision Letter 1]

3 Jan 2025

PONE-D-24-49440R1The Impact of Physical Activity Variety on Physical Activity ParticipationPLOS ONE

Dear Dr. Dregney,

Thank you for submitting your manuscript to PLOS ONE. After careful consideration, we feel that it has merit but does not fully meet PLOS ONE’s publication criteria as it currently stands. Therefore, we invite you to submit a revised version of the manuscript that addresses the points raised during the review process.

We look forward to receiving your revised manuscript.

Kind regards,

Henri Tilga, PhD

Academic Editor

PLOS ONE

Journal Requirements:

Reviewers' comments:

Reviewer's Responses to Questions

**Comments to the Author**

1. If the authors have adequately addressed your comments raised in a previous round of review and you feel that this manuscript is now acceptable for publication, you may indicate that here to bypass the “Comments to the Author” section, enter your conflict of interest statement in the “Confidential to Editor” section, and submit your "Accept" recommendation.

Reviewer #1: (No Response)

2. Is the manuscript technically sound, and do the data support the conclusions?

Reviewer #1: Yes

3. Has the statistical analysis been performed appropriately and rigorously? 

Reviewer #1: Yes

4. Have the authors made all data underlying the findings in their manuscript fully available?

Reviewer #1: No

5. Is the manuscript presented in an intelligible fashion and written in standard English?

Reviewer #1: Yes

6. Review Comments to the Author

Reviewer #1: Abstract

1. The abstract mentions “marginally significant” results (e.g., p = 0.072), which may be misleading for readers. Revise this to reflect the exploratory nature of these findings.

2. The description of the intervention and major outcomes in the abstract should be more concise while retaining clarity.

Introduction

In discussing PA variety, the authors cite limited research but fail to integrate all relevant studies fully. Include additional literature.

Methods

1. Provide more details regarding the ActiGraph device’s reliability and thresholds for defining valid wear-time and MVPA.

2. The counseling sessions’ structure is described, but specific motivational strategies (e.g., goal setting, feedback) should be outlined more clearly.

Results

Some tables (e.g., psychosocial variables) are detailed but dense. Consider condensing tables into key findings and relegating supplementary data to appendices.

Discussion

1. Expand on the finding that the consistency group also improved in MVPA and motivation. This suggests that weekly counseling sessions might have driven outcomes, not just the variety intervention. Discuss this as a potential confounder.

2. Highlight the implications for practitioners more clearly, focusing on the feasibility of home-based variety interventions and strategies to enhance adherence.

Major Concerns:

1. While the authors report significant increases in variables such as self-efficacy, autonomy, and tranquility, many findings are described as “marginally significant” (p-values near 0.10). It would be helpful to reframe these results cautiously and avoid overstating conclusions. Additionally, providing effect sizes for all analyses would clarify the magnitude of observed differences.

2. The authors report adherence rates and satisfaction scores, which are commendable. However, further details on how participants adhered to completing varied workouts (e.g., participant logs or tracking systems) would strengthen the feasibility findings.

7. PLOS authors have the option to publish the peer review history of their article (what does this mean? ). If published, this will include your full peer review and any attached files.

**Do you want your identity to be public for this peer review?** For information about this choice, including consent withdrawal, please see our Privacy Policy .

Reviewer #1: **Yes: ** Sarieh Poortaghi

---

## [Author Response · Author response to Decision Letter 1]

14 Feb 2025

Journal Requirements

a. Thank you for pointing this out. We have corrected a mistake in that citations 47 and 48 were listed in the incorrect order. This mistake has been fixed on the reference page. Additionally, the citation at the bottom of page 3 has been updated to reflect the reference page with the article from Glaros & Janelle cited as article #20.

Reviewer 1

1. The abstract mentions “marginally significant” results (e.g., p = 0.072), which may be misleading for readers. Revise this to reflect the exploratory nature of these findings.

a. We agree with this point and appreciate the note. The methods and discussion portion of the abstract has been modified to reflect the exploratory nature of this study (see abstract).

2. The description of the intervention and major outcomes in the abstract should be more concise while retaining clarity.

a. We appreciate you pointing this out. We have cut down both the intervention description and the findings portion of the abstract.

3. In discussing PA variety, the authors cite limited research but fail to integrate all relevant studies fully. Include additional literature.

a. Thank you for pointing this out. We have added more research to the introduction at the bottom of page 3. Additional citations have also been added in the first paragraph of page 4.

4. Provide more details regarding the ActiGraph device’s reliability and thresholds for defining valid wear-time and MVPA.

a. We appreciate you for raising this concern and agree with the sentiment. We have added further discussion regarding the two day threshold, why it was used, and what could have been done differently in the limitations section (see paragraph 2, page 19).

5. The counseling sessions’ structure is described, but specific motivational strategies (e.g., goal setting, feedback) should be outlined more clearly.

a. We agree with this note. A table has been added to pages 7 and 8 that details the individual topics of each session.

6. Some tables (e.g., psychosocial variables) are detailed but dense. Consider condensing tables into key findings and relegating supplementary data to appendices.

a. We appreciate you pointing this out. We have modified the PA table to now be two supplementary tables to make it more concise and provide clarity (see supplemental tables 1 and 2).

7. Expand on the finding that the consistency group also improved in MVPA and motivation. This suggests that weekly counseling sessions might have driven outcomes, not just the variety intervention. Discuss this as a potential confounder.

a. Thank you for pointing this out. More detail has been added to the first paragraph of page 18.

8. Highlight the implications for practitioners more clearly, focusing on the feasibility of home-based variety interventions and strategies to enhance adherence.

a. We agree with this sentiment. A section has been added to the final paragraph on page 21 addressing these concerns.

9. While the authors report significant increases in variables such as self-efficacy, autonomy, and tranquility, many findings are described as “marginally significant” (p-values near 0.10). It would be helpful to reframe these results cautiously and avoid overstating conclusions. Additionally, providing effect sizes for all analyses would clarify the magnitude of observed differences.

a. Thank you for raising this concern. We have tried to be intentional about using the term “marginal” when discussing these findings to indicate that findings be used cautiously whenever possible. Additionally, we have added to the limitations section to clearly state the concerns you have raised (see page 20, paragraph 1).

b. Additionally, effect sizes have been added to all relevant findings.

10. The authors report adherence rates and satisfaction scores, which are commendable. However, further details on how participants adhered to completing varied workouts (e.g., participant logs or tracking systems) would strengthen the feasibility findings.

a. We apologize for not having this included earlier. We have now added a description of how activity was tracked at the bottom of page 8 and top of page 9.

---

## [Decision Letter · Decision Letter 2]

7 Mar 2025

PONE-D-24-49440R2The Impact of Physical Activity Variety on Physical Activity ParticipationPLOS ONE

Dear Dr. Dregney,

Thank you for submitting your manuscript to PLOS ONE. After careful consideration, we feel that it has merit but does not fully meet PLOS ONE’s publication criteria as it currently stands. Therefore, we invite you to submit a revised version of the manuscript that addresses the points raised during the review process.

We look forward to receiving your revised manuscript.

Kind regards,

Henri Tilga, PhD

Academic Editor

PLOS ONE

Reviewers' comments:

Reviewer's Responses to Questions

**Comments to the Author**

1. If the authors have adequately addressed your comments raised in a previous round of review and you feel that this manuscript is now acceptable for publication, you may indicate that here to bypass the “Comments to the Author” section, enter your conflict of interest statement in the “Confidential to Editor” section, and submit your "Accept" recommendation.

Reviewer #1: (No Response)

2. Is the manuscript technically sound, and do the data support the conclusions?

Reviewer #1: No

3. Has the statistical analysis been performed appropriately and rigorously? 

Reviewer #1: No

4. Have the authors made all data underlying the findings in their manuscript fully available?

Reviewer #1: No

5. Is the manuscript presented in an intelligible fashion and written in standard English?

Reviewer #1: Yes

6. Review Comments to the Author

Reviewer #1: There are several areas requiring improvement, particularly in methodology clarity, statistical interpretation, and discussion depth.

Abstract:

The use of "marginally significant" (p = 0.072) is misleading. Results should be cautiously interpreted.

The conclusion should briefly suggest practical applications of the findings.

Introduction

Some references are outdated (e.g., pre-2010 studies). Recent research (2022-2024) should be included.

The research gap should be more explicitly stated.

Methods

Sample size (n = 47) is small, reducing generalizability.

Lack of power analysis to justify sample size.

Weekly counseling sessions for both groups could be a confounder—this should be discussed in limitations.

Results

Many findings are described as "marginally significant" (p-values near 0.10). These should be cautiously framed.

Effect sizes should be reported consistently.

No post-hoc tests or multiple comparison corrections were mentioned.

Discussion

Does not critically assess why some hypotheses were not supported.

The role of PA enjoyment and boredom reduction needs deeper exploration.

Future research directions should be more specific.

Conclusion

Practical applications are vague—how can practitioners use these findings?

Should clearly discuss policy or intervention implications.

References

Some references are outdated (pre-2010).

A few recent studies (2022-2024) should be included to reflect current trends.

7. PLOS authors have the option to publish the peer review history of their article (what does this mean? ). If published, this will include your full peer review and any attached files.

**Do you want your identity to be public for this peer review?** For information about this choice, including consent withdrawal, please see our Privacy Policy .

Reviewer #1: **Yes: ** Sarieh Poortaghi

---

## [Author Response · Author response to Decision Letter 2]

20 Mar 2025

Reviewer 1

Abstract:

1. The use of "marginally significant" (p = 0.072) is misleading. Results should be cautiously interpreted.

a. Thank you for raising this point. In the conclusion section of the abstract, there is now a disclaimer to interpret findings with caution given the use of marginal significance (see abstract).

2. The conclusion should briefly suggest practical applications of the findings.

a. We appreciate you raising this point. The conclusion of the abstract now concludes with practical applications of the findings related to supporting client psychological needs and providing home-based physical activity options (see abstract).

Introduction

3. Some references are outdated (e.g., pre-2010 studies). Recent research (2022-2024) should be included.

a. Thank you pointing this out. Study citation #4, 13, and 28 have been updated with more recent sources that provide updated information on the topic. Due to the limited number of studies that examined variety in physical activity program previously, authors deemed studies discussing this topic necessary to keep as they help set up the research gap.

4. The research gap should be more explicitly stated.

a. We feel this is a great point. We have gone back to the introduction to clarify the gaps this project was addressing and wrote them as a bridge to the purpose of the study (see top of page 5).

Methods

5. Sample size (n = 47) is small, reducing generalizability.

a. Thank you for raising this point. In the limitations section of the discussion, we have added more on why we used a small sample size. The reason being this study was a feasibility and acceptability study (see page 20, first paragraph).

6. Lack of power analysis to justify sample size.

a. Following the previous point, we also discussed why the small sample size used in a feasibility and acceptability study meant it would not be appropriate to have a power analysis. This is discussed as a limitation (see page 20, first paragraph).

7. Weekly counseling sessions for both groups could be a confounder—this should be discussed in limitations.

a. We appreciate this point. We have added a sentence discussing this confounder to the discussion section (see page 20, second paragraph).

Results

8. Many findings are described as "marginally significant" (p-values near 0.10). These should be cautiously framed.

a. We agree with this point and have addressed this issue in the abstract and the limitations section to clarify why this term was used (see abstract and page 20, first paragraph)

9. Effect sizes should be reported consistently.

a. Thank you for catching this. We have no added effect sizes for significant findings throughout the document.

10. No post-hoc tests or multiple comparison corrections were mentioned.

a. We appreciate you raising this concern. However, due to the small sample size and exploratory nature of this study, post-hoc tests were not used to avoid the risk of over misinterpretation.

Discussion

11. Does not critically assess why some hypotheses were not supported.

a. We agree with this concern. Discussion has been added to various sections of the discussion section where an unsupported hypothesis is discussed (see pages 18-19).

12. The role of PA enjoyment and boredom reduction needs deeper exploration.

a. Thank you for raising this point. Further clarification has been added to this discussion to talk about how enjoyment and boredom relate to other components and findings in this study (see page 19, paragraph 2).

13. Future research directions should be more specific.

a. We appreciate this point being brought to our attention. We have clarified some specific strategies in which future research directions could move (see pages 21-22, final paragraph-first paragraph).

Conclusion

14. Practical applications are vague—how can practitioners use these findings?

a. Thank you for raising this concern. The practical applications have been clarified (see end of page 22).

15. Should clearly discuss policy or intervention implications.

a. Thank you for raising this. We feel that the clarification of intervention implications in the discussion section has provided clarity to the discussion of this during the conclusion.

References

16. Some references are outdated (pre-2010).

17. A few recent studies (2022-2024) should be included to reflect current trends.

a. To address both of these points, where suitable, older studies were replaced with more recent ones to reflect the current environment. However, do the limited nature of studies similar to the present one, some integral older studies are still featured.

---

## [Editor Report · Decision Letter 3]

4 Apr 2025

The Impact of Physical Activity Variety on Physical Activity Participation

PONE-D-24-49440R3

Dear Dr. Dregney,

We’re pleased to inform you that your manuscript has been judged scientifically suitable for publication and will be formally accepted for publication once it meets all outstanding technical requirements.

Kind regards,

Henri Tilga, PhD

Academic Editor

PLOS ONE
---

## [Editor Report · Acceptance letter]

PONE-D-24-49440R3

PLOS ONE

Dear Dr. Dregney,

I'm pleased to inform you that your manuscript has been deemed suitable for publication in PLOS ONE. Congratulations! Your manuscript is now being handed over to our production team.

Kind regards,

on behalf of

Dr. Henri Tilga

Academic Editor

PLOS ONE